# Antibacterial Calcium Phosphate Coatings with Zinc Oxide Nanoparticles

**Valentina Chebodaeva** [1,2,*], **Mariya Sedelnikova** [1], **Margarita Khimich** [1], **Olga Bakina** [1], **Alexey Tolmachev** [1], **Andrey Miller** [1], **Kirill Golohvast** [3], **Aleksander Zakharenko** [3], **Vladimir Egorkin** [4], **Igor Vyaliy** [4] **and Yurii Sharkeev** [1]

1   Institute of Strength Physics and Materials Science, Siberian Branch of the Russian Academy of Sciences, 2/4 Academicheskii Pr., Tomsk 634055, Russia
2   Laboratory of Cellular and Microfluidic Technologies, Siberian State Medical University, Tomsk 634050, Russia
3   Research Centers and Laboratories Nanotechnology, School of Engineering, Far Eastern Federal University, Sukhanova Str. 8, Vladivostok 690090, Russia
4   Institute of Chemistry, Far Eastern Branch of the Russian Academy of Sciences, Vladivostok 690022, Russia
*   Correspondence: valentinoch@ispms.ru

**Abstract:** Porous calcium phosphate coatings (C-P) with ZnO nanoparticles were obtained via the micro-arc oxidation method on a titanium substrate. ZnO nanoparticles were added to the C-P coatings to change the zeta potential and improve the coatings' bioactivity and antibacterial properties. The samples with coatings were studied via scanning electron microscopy (SEM), X-ray diffraction, energy dispersive microanalysis, potentiodynamic polarization, and zeta potential measurement. The coatings modified with ZnO nanoparticles showed improved physical, electrochemical, and electrical properties, compared to the initial unmodified coatings. Modification with ZnO nanoparticles contributed to an increase in zeta potential from $-60$ mV to $-53$ mV. Functionalization of the coatings with ZnO nanoparticles allowed us to increase the anticorrosion characteristics by about 30%. The biological studies showed that the coatings had no cytotoxic effect on L929 fibroblast cells. The antibacterial activity of the coating rose by 99% after the addition of ZnO nanoparticles against the bacterium *Staphylococcus aureus.*

**Keywords:** calcium phosphate coatings; micro-arc oxidation; nanoparticles; zeta potential; electrical properties; antibacterial activity

## 1. Introduction

Biocompatible metallic implants for osseointegration have been successfully used in orthopedic surgery for many years [1–4]. However, rejection of the implants and inflammation in the surrounding bone tissue is observed in 10–30% of surgical cases [5]. Currently, the following metals and alloys are used as materials for the production of bone implants: zirconium, tantalum, stainless steel, chromium–cobalt alloys, silver–palladium alloys, magnesium, and other materials [6–9]. In most cases, titanium and titanium-based alloys are used for bone implants due to their mechanical properties, high strength, and biocompatibility [2,10,11]. Implant properties such as the surface topography, charge, corrosion resistance, and chemical composition are important for the successful ingrowth of material into bone tissue [12–16]. Rough and porous surfaces have a more pronounced and positive effect on cell activity compared to smooth surfaces [9,17,18]. There are various methods for treating the surface of metals to increase their biocompatibility, for example, surface alloying, thermal, or deformation treatment [19,20]. An effective way to create the required chemical composition and roughness of the implant surface is the formation of a bioactive calcium phosphate (C-P) coating on the metal substrate [21–24]. In addition to the surface topography, the implant surface charge influences cell adhesion. It follows from [25,26] that charged surfaces increase osteoconductivity and contribute to bone tissue repair.

Plasma electrolytic oxidation (PEO) or micro-arc oxidation (MAO) allows the formation of calcium phosphate and oxide coatings on such metals as Ti, Al, Mg, Zr, and their alloys to improve their bioactivity and corrosion resistance [10,16,23]. In addition, the high porosity of MAO coatings allows them to be saturated with various antibacterial and healing agents. During the deposition process, a certain electrical state is formed on the coatings' surface. The charge of such coatings is negative due to the positive charge applied to the substrate [24]. Opposite-sign ions from the electrolyte are deposited on the surface of the substrate. Since the human cell membrane is negatively charged [10,25], it makes sense to add a positive charge to the implant surface to attract bone cells to its surface.

In recent decades, intensive research has been conducted to develop an inorganic two-dimensional nanostructure with controlled drug delivery [27,28]. Nanoscale particles have a high surface energy due to their small size, which provides their high chemical activity. Moreover, ZnO nanostructures have proven antibacterial properties and are widely used as additives for disinfection [29,30]. The use of ZnO nanoparticles as an antibacterial agent is promising, since they do not form resistant strains of microorganisms, unlike antibiotics. This will allow us to avoid the risks of wound infection and the creation of resistant bacteria at the implantation site. In addition, ZnO nanoparticles obtained via the method of electric explosion have an electropositive charge in aqueous media, which allows them to change the electrical potential of a biocoating's surface [27]. The introduction of such nanoparticles into a C-P coating changes its charge state and antibacterial characteristics. The modification of biomaterials with such nanoparticles will improve their bioactive, electrical, and chemical properties. Currently, many researchers are introducing nano- and microparticles into coatings for additional functionalization [14,30–33]. For example, the authors of [15] introduced ZnO–ZrO$_2$ nanoparticles into TiO$_2$ coatings' composition to increase the corrosion resistance and antibacterial properties of the coatings. And in [30], titanium surfaces were successfully modified using zinc-containing nanowires, which made it possible to enhance cell adhesion, proliferation, and osteogenic differentiation.

This field is being actively developed, and further work carried out in this area will significantly expand our understanding of the mechanisms of influence of the introduction of such additives on materials' properties.

This work aimed at the formation of bactericidal coatings containing ZnO nanoparticles in order to change the structural, electrical, antibacterial, and other properties of the coatings.

## 2. Materials and Methods

### 2.1. Sample Characterization and MAO Preparation

Commercially pure titanium (ASTM Grade 2) plates ($1 \times 10 \times 10$ mm$^3$) (VSMPO-AVISMA Corp., Verkhnaya Salda, Russia) were used in this study. A stoichiometric hydroxyapatite, Ca$_{10}$(PO$_4$)$_6$(OH)$_2$ (ISSCM SB RAS, Novosibirsk, Russia), obtained as a powder via the method of mechanochemical synthesis, was used for the coating formation [34]. Powders of CaCO$_3$ and H$_3$PO$_4$ water solution were used for the electrolyte production, as described in [9,12,17]. The "Microarc–3.0" installation developed at the Institute of Strength Physics and Materials Science SB RAS (ISPMS SB RAS, Tomsk, Russia) was used to carry out the MAO (PEO) process. The coating formation process was carried out in anodic potentiostatic mode at a pulse frequency of 50 Hz, pulse duration of 100 µs, and applied voltage of 200 V for 10 min. The ZnO nanopowder was produced via electrical explosion of a zinc wire in an argon (80%) and oxygen (20%) atmosphere [27].

The ZnO nanoparticles were applied to the C-P coatings using impregnation and assisted ultrasonic (US) dispersion in 25 mL of distilled water. An ultrasonic bath ("Sapphire") was used for the ultrasonic dispersion. The ultrasonic frequency and power were 35 kHz and 100 W, respectively. A general scheme of the coatings' modification using ZnO nanoparticles is shown in Figure 1.

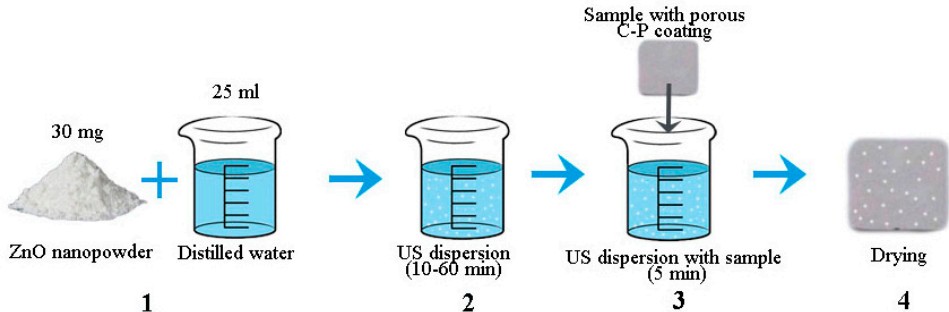

**Figure 1.** General scheme of coating modification using ZnO nanoparticles.

Structural and elemental analyses of the coatings were carried out using a LEO EVO 50 scanning electron microscope (Zeiss, Jena, Germany) with an attachment for energy dispersive analysis. The phase composition of the coated samples was identified by means of X-ray diffraction analysis (DRON 8N (Bourevestnik, St. Petersburg, Russia) equipped with a Mythen 2R 1D microstrip X-ray detector (Dectris, Hyogo, Japan)). All the measurements were performed with Bragg–Brentano geometry in Cu K$\alpha$ radiation ($\lambda = 0.15406$ nm) within a 2$\theta$ angle range of 10–90°. The detector scanning step was 0.1°, and the acquisition time was 10 s. The distribution of ZnO nanopowder particles was studied via sedimentation analysis using a CPS DC24000 UHR centrifuge (Analytik Ltd., Cambridge, UK). For the experiments, powder suspensions, 0.1 mL each, were prepared under vigorous stirring. The calibration solution introduced into the disk included particles with a size of 0.377 or 0.460 $\mu$m. The zeta potential of the coatings was measured using a Z-potentiometer (Anton Paar SurPASS, Ostfildern-Scharnhausen, Germany) in an aqueous solution of KCl (0.05 mol/L) at the Nanotechnology Research and Education Center, FEFU (Vladivostok, Russia). Measurements of the zeta potential of coatings obtained at different durations of dispersion of suspensions were carried out on at least 3 samples of each group. To calculate the surface porosity of coatings from SEM images, the standard "secant" method was used [13,17]. The basic principle of this method is that the fraction of the phase (pores) in the volume of the coating is equal to the part of the secant line passing through this phase (pores) on the area of the image [13,17].

The Versa STAT MC system (Princeton Applied Research, Oak Ridge, TN, USA) was used to investigate the corrosion properties of the C-P coatings on titanium substrates. Measurements were conducted in a three-electrode cell in Ringer solution. The Ringer solution temperature during the experiment was 37 °C.

*2.2. Investigation of Biological Properties*

The cytotoxicity of the C-P coatings was studied by means of MTT testing using the mouse fibroblast L929 cell line (Russian Research Center of Virology and Biotechnology VECTOR, Russia). This cell line is recommended by ISO 10993-5 due to its high sensitivity to toxic effects. The cells were cultured within 24 h in Dulbecco's Modified Eagle Medium (HiClone, Logan, UT, USA) with the addition of 10% fetal bovine serum (HiClone) and 1% penicillin/streptomycin (Biolot, St. Petersburg, Russia) at 37 °C and 5% CO$_2$ (Sanyo, Tokyo, Japan). For cytotoxicity investigation via the extract test, test samples with 2 mL DMEM were placed in the wells of a 24-well plate and incubated for 24 h at 37 °C. The samples were taken out and a cell suspension was added in the amount of 50,000 cells per sample. The cells were incubated for 24 h at 37 °C in an atmosphere of 5% CO$_2$. After incubation, cells were detached from the surface of the composite samples and transferred as 100 $\mu$L cell suspensions each into 96-well plates. Then, 10 $\mu$L of MTT solution (3-(4.5-dimethyl thiazol-2-yl)-2.5-diphenyl-tetrazolium bromide) was added to each well. The cells were incubated with MTT solution at 37 $\pm$ 1 °C for 2 h in an atmosphere of 5% CO$_2$. At the end of the incubation period, the nutrient medium was carefully removed, and 100 $\mu$L of dimethyl sulfoxide (Biolot, Russia) was added to each well to dissolve formazan

crystals. After 15 min, the optical density was determined using a Multiscan FC microplate spectrophotometer (Termo Scientific, Berlin, Germany) at a wavelength of 570 nm. Then, the percentage of living cells (viability, %) was calculated.

The antibacterial activity of the C-P coating samples was determined by counting the residual bacteria using the CFU counting method against methicillin-resistant *Staphylococcus aureus* (MRSA) hospital strain No. 209. MRSA was cultured at 37 °C in Mueller–Hinton agar (Merck, Rahway, NJ, USA) for 24 h. The test samples were placed in Petri dishes with a diameter of 60 mm. A volume of 1 μL per 1 mm$^2$ sample area of the bacterial suspension in Mueller–Hinton broth (HiMedia Laboratories, Maharashtra, India) with a concentration of $10^5$ CFU/mL was placed on the sample surfaces and spread evenly using a sterile swab. The exposure time was 3 or 6 h at 37 °C and a humidity of 90%. The residual number of viable bacteria (CFU/mL) was counted after 3 and 6 h of incubation at 37 °C. After exposure, samples were placed in sterile containers containing 10 mL of neutralizing agent (PBS, HiMedia Laboratories, Maharashtra, India) and shaken for 10 min using a PSU-20i orbital shaker (SIA Biosan, Riga, Latvia) to remove adhered bacteria. From each supernatant, 100 μL was placed onto Mueller–Hinton agar plates. The prepared plates were subsequently incubated at 37 ± 1 °C for 24 h. The colony plate count method was used for the enumeration of CFU after incubation. The antibacterial rate (AR) of each sample was evaluated by calculating the percent (%) reduction in microbial contamination of the test samples compared to the control samples. Bacteria in PBS only were used as control samples.

For each of the samples, two independent experiments were conducted, with five repetitions per sample. Statistical analysis was carried out using the unpaired Student's *t* test, while *p* < 0.05 was deemed statistically significant.

## 3. Results and Discussion

### 3.1. Morphology of the C-P Coatings

Figure 2 presents a TEM image and the particle size distribution of the ZnO metal nanopowder. The ZnO particle distribution has a monomodal character, and the average particle size was 50 ± 2 nm. The powder contained particles larger than 100 nm, but their volume fraction did not exceed 15%. An increase in the duration of US of the nanopowder suspension from 10 to 60 min led to an increase in the fraction of particles with a size of 20–50 nm from 77 to 90%. Further increase in the duration of US up to 70 min reduced the fraction of ZnO particles with a size of 20–50 nm down to 75%.

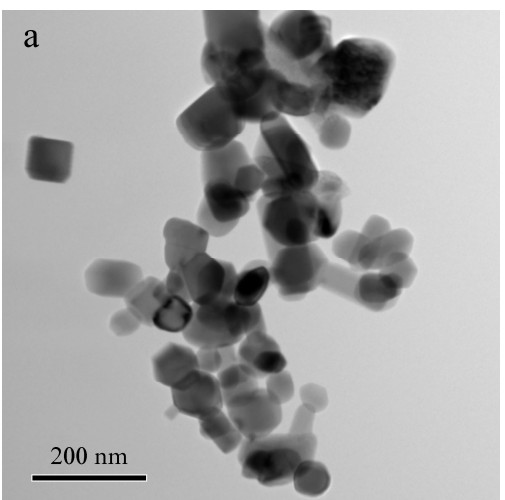
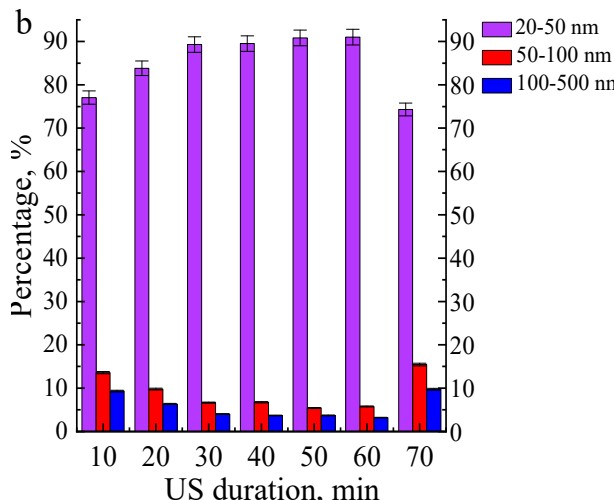

**Figure 2.** TEM image of ZnO nanopowder (**a**) and size distribution of the ZnO nanoparticle agglomerates depending on the ultrasonic dispersion duration (10–60 min) (**b**).

The ZnO nanoparticle fraction with sizes in the range of 20–50 nm is significantly higher compared to the nanoparticle fraction with larger sizes. The nanoparticles' shape and structure contribute to reducing the attraction forces between particles. As shown in [24,35], AlO(OH) nanoparticles after hydrolysis are characterized by a complex structure with needle branches, which leads to a high binding energy between nanoparticles and high fractions of particles with larger sizes. In contrast, the ZnO nanoparticles have geometric shapes (hexagon, circle, square, rectangle) (Figure 2a), which can reduce particle agglomeration.

Figure 3 represents SEM images of the coating surface before and after the addition of ZnO nanopowder. The surface of all samples is characterized by the presence of spherical porous formations. This relief is characteristic of the morphology obtained via the MAO method and is due to the processes of microarc discharges and electric breakdowns occurring on the metal substrate [17,21,24]. US dispersion of the aqueous suspension of ZnO nanopowder for 10 min and sedimentation of ZnO nanoparticles on the C-P coating surface led to the formation of dense spherical particles with sizes of up to 8 μm and a few smaller agglomerates of ZnO nanoparticles (1–2 μm) (Figure 3b). The roughness (Ra) of this coating was 3.4 μm (Figure 4a).

An increase in the duration of US dispersion of the ZnO suspension led to a gradual decrease in the ZnO agglomerates' sizes from 4.5 to 1.5 μm, along with a more uniform distribution over the coating surface (Figures 3c–f and 4b).

With increasing duration of US dispersion of the ZnO suspension, the porosity of the coating linearly increased from 33 to 45% due to the partial breakdown of the coating structural elements (Figure 4c). This led to a growth in the roughness of the coatings from 3.4 to 4.7 μm with an increase in the US dispersion duration from 10 to 60 min.

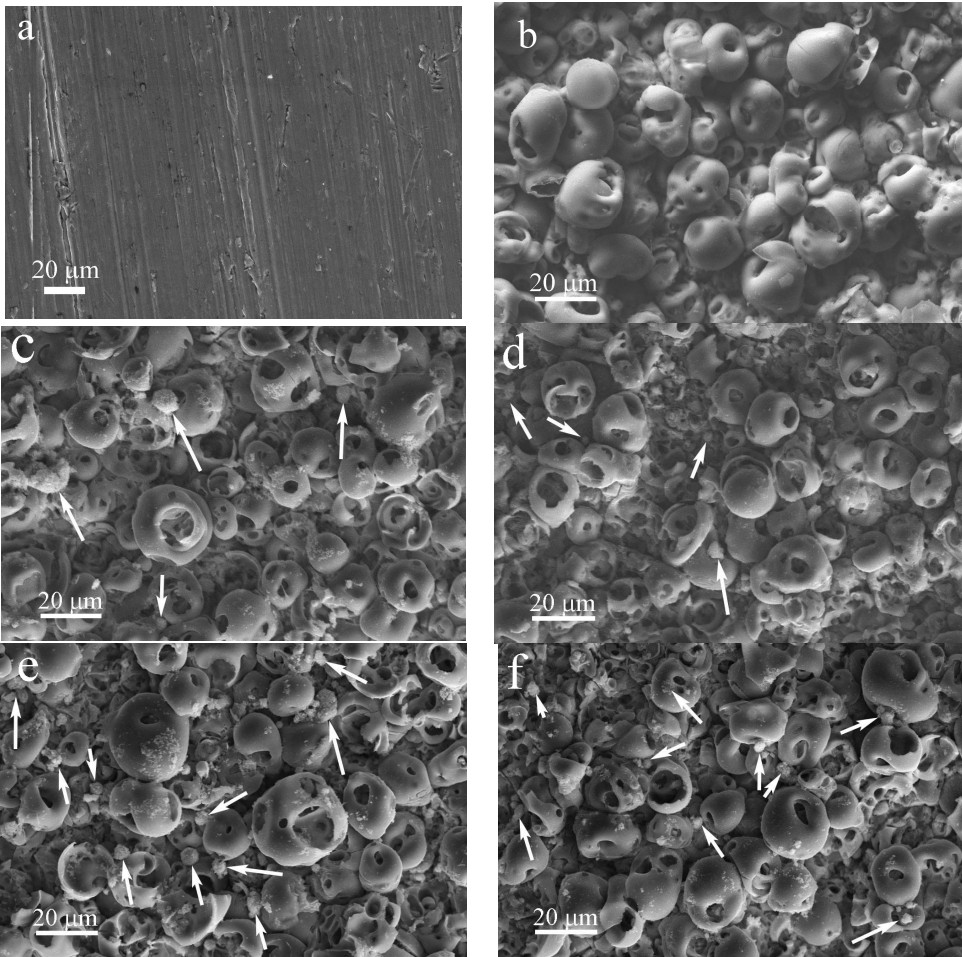

**Figure 3.** *Cont.*

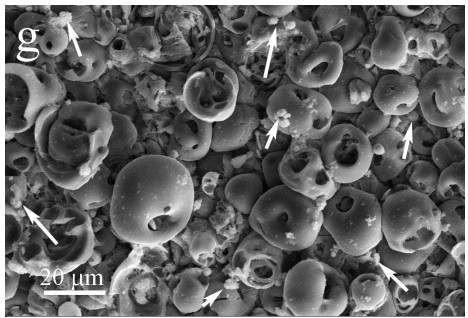
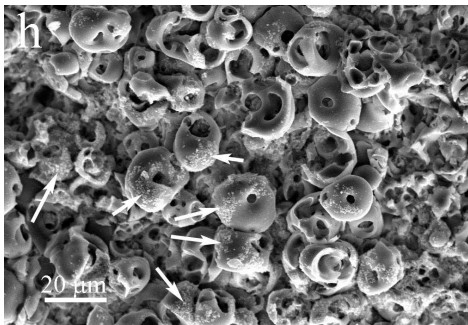

**Figure 3.** SEM images of the titanium substrate (**a**) and coating surface before (**b**) and after production with the addition of ZnO nanopowder after US dispersion of ZnO suspension for different durations, min: (**c**) 10, (**d**) 20, (**e**) 30, (**f**) 40, (**g**) 50, (**h**) 60 (arrows point to the agglomerates of ZnO nanoparticles.).

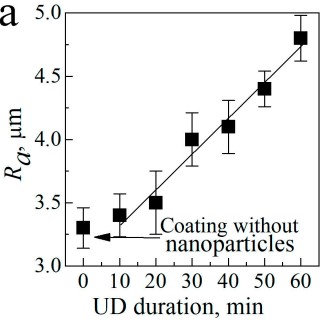
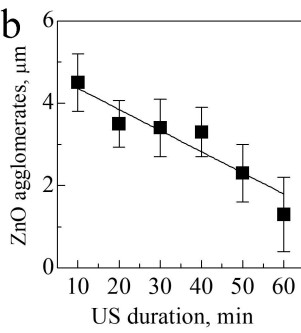
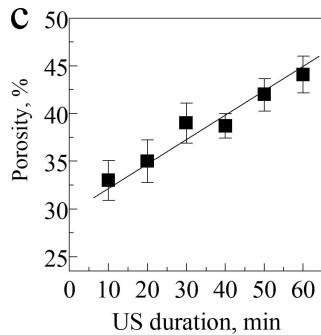

**Figure 4.** Dependence of the roughness (**a**), size of ZnO agglomerates (**b**), and porosity (**c**) of the coatings on the US duration of the ZnO suspension.

### 3.2. Elemental and Phase Composition of Functionalized Coatings

Figure 5 shows SEM images and elemental maps of the coatings modified with ZnO nanoparticles after different durations (20, 40, and 60 min) of US dispersion of the ZnO nanopowder. It can be seen that calcium and phosphorus were evenly distributed over the surface of the coating, whereas zinc was mainly concentrated in small particles at the coating surface (Figure 5).

An increase in the US duration of the ZnO suspension led to a decrease in the ZnO nanoparticle size of the coatings and a more homogeneous distribution of ZnO nanoparticles over the coating surface.

The elemental composition of the C-P coatings containing ZnO nanoparticles is presented in Table 1. There was a high content of elements such as O (66–69 at. %), P (14–15 at. %), and Ti (11–12 at. %), while Ca (3–4 at. %) and Zn (1–3 at. %) were present in small amounts (Table 1).

**Table 1.** Elemental composition of the coatings with ZnO nanoparticles, at. %.

| Elements | US Duration | | | | | |
| --- | --- | --- | --- | --- | --- | --- |
| | 10 | 20 | 30 | 40 | 50 | 60 |
| Zn | 3 | 3 | 2 | 2 | 2 | 1 |
| O | 66 | 67 | 69 | 66 | 69 | 69 |
| P | 15 | 14 | 15 | 15 | 14 | 15 |
| Ca | 4 | 4 | 3 | 4 | 4 | 4 |
| Ti | 12 | 12 | 11 | 12 | 11 | 11 |
| Ca/P | 0.3 | 0.3 | 0.3 | 0.3 | 0.3 | 0.3 |

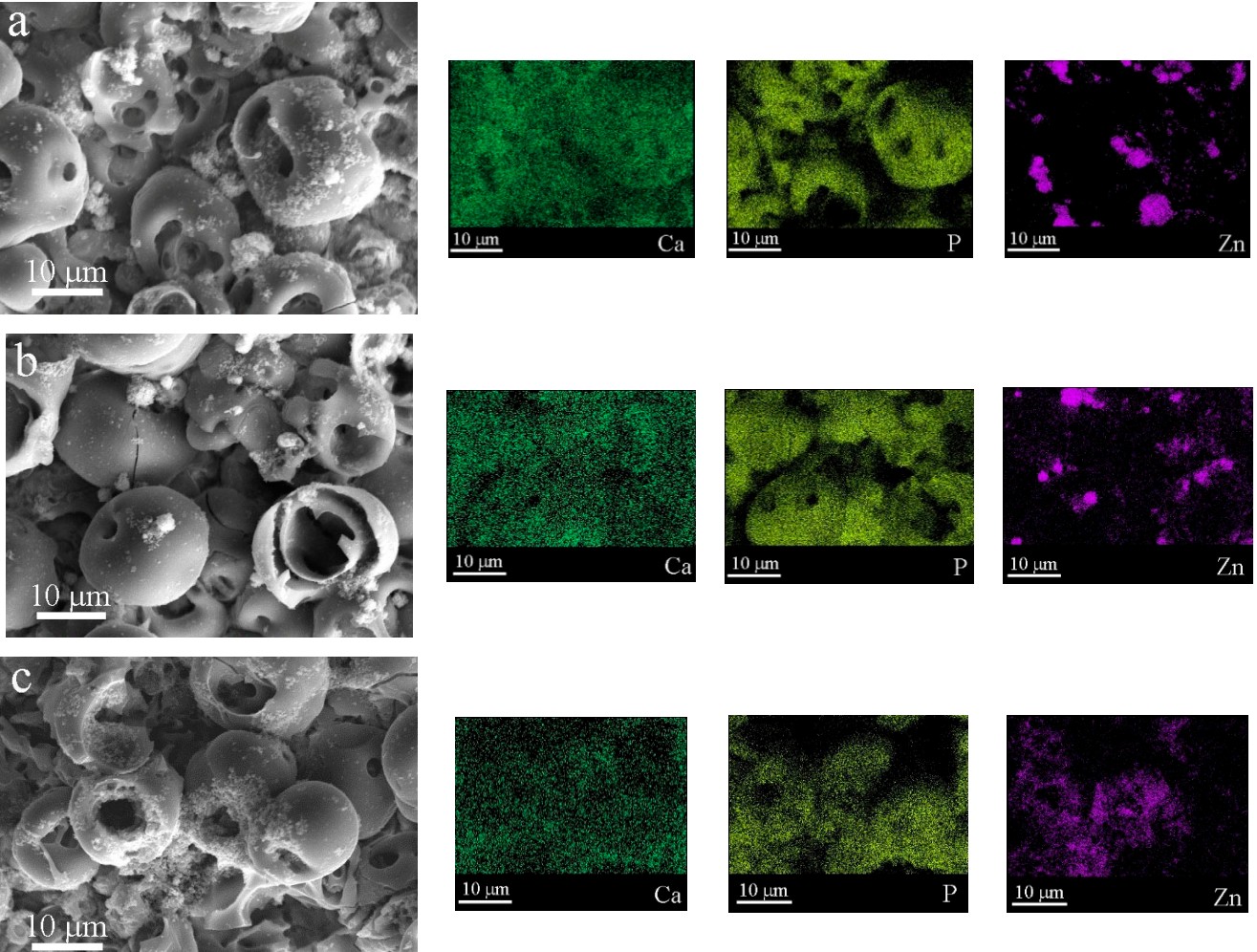

**Figure 5.** SEM images of C-P coatings and distribution maps of Ca, P, and Zn in the coatings produced under different US durations of ZnO suspension, min: (**a**) 20, (**b**) 40, (**c**) 60.

The Ca/P ratio for the modified C-P coatings was 0.3. As the US duration of the ZnO suspension decreased from 60 to 10 min, the Zn content decreased from 3 to 1%. This resulted in a decrease in the agglomerate sizes and ZnO nanoparticle penetration into the inner volume of the coating.

Since the most homogeneous distribution of the ZnO nanoparticles was observed for the coating modified after 60 min of dispersion, a cross section of this coating was studied (Figure 6). The elemental composition in the cross section of the coating confirmed that ZnO entered the pores of the coating (Figure 6d).

It should also be noted that the Zn was present across the entire coating. In some areas of the coating, the concentration of particles was high. Ca and P were evenly distributed throughout the coating. The SEM images of the cross section of the coatings showed a highly porous structure, with an average pore size of 3.5 mm (Figure 6).

Figure 7 shows XRD patterns of the C-P coatings before and after modification with ZnO suspension after 60 min of US dispersion. The coatings were X-ray amorphous, which was confirmed by the presence of a diffusive scattering area at small angles ranging from 20 to 45 degrees (Figure 7). There were several peaks corresponding to the ZnO phase in the coating patterns.

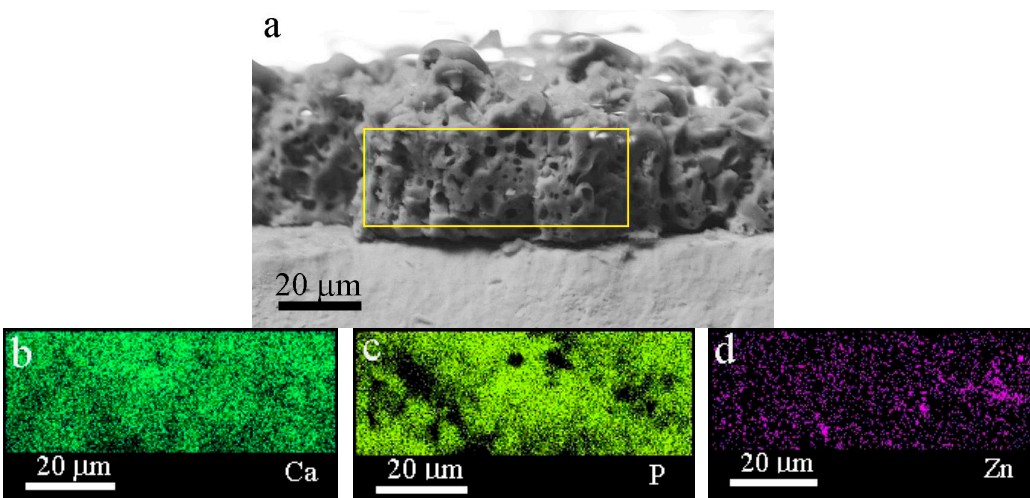

**Figure 6.** SEM image (**a**) and maps (**b**–**d**) of a cross section of the coating produced using ZnO nanoparticles after 60 min of US dispersion.

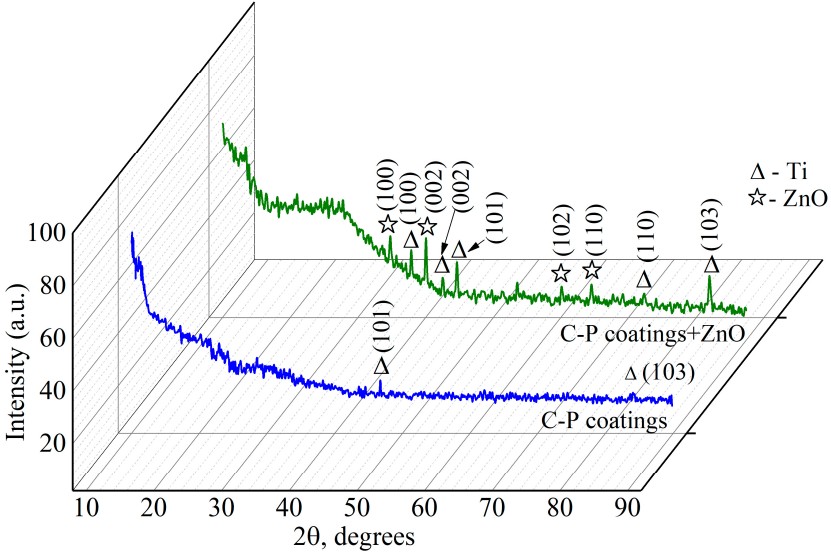

**Figure 7.** XRD patterns of C-P coatings before and after the addition of ZnO nanoparticles after 60 min of US dispersion.

The XRD patterns of the C-P coatings show reflexes that refer to the titanium substrate (Ti). It can be seen from the X-ray diffraction patterns that there were more substrate peaks after ZnO nanoparticle addition. This is due to the partial destruction of the spherical coating formations and the occurrence of channel coating pores.

### 3.3. Electrical Properties of Modified Coatings

Figure 8 illustrates the zeta potential of the C-P coatings without ZnO nanoparticles and with ZnO nanopowder after different durations of US dispersion. The coatings had a negative potential, ranging from −63 mV to −53 mV.

This result is due to the formation of the coating using the MAO method in the anodic mode. In this case, the Ti substrate acts as an anode and has a positive potential. Negative ions flow onto the Ti substrate and form a negatively charged coating [7,22,24–26].

The homogeneously distributed positively charged ZnO nanoparticles led to the highest zeta potential among all the coatings (Figure 8).

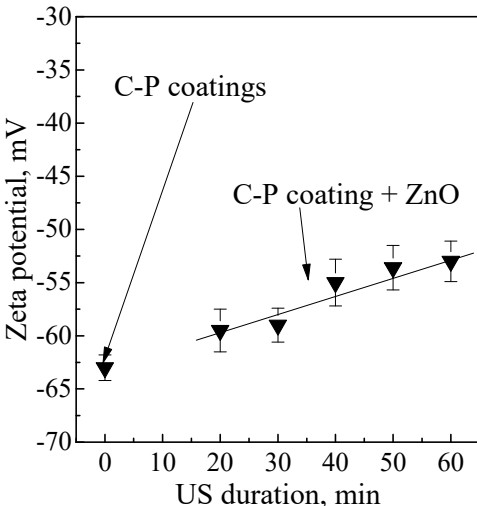

**Figure 8.** Dependence of the zeta potential value of C-P coatings on the duration of US dispersion before and after modification of coatings with ZnO nanoparticles.

An increase in the duration of US from 10 to 20 min led to a linear increase in the zeta potential value from −60 mV to −53 mV. This increase in the zeta potential value of the coating surface was caused by partial compensation of the coatings' negative charge by the positively charged nanoparticles.

Figure 9 shows a scheme of the electrical charge distribution on the surface of MAO coatings without ZnO nanoparticles (a, b) and after different durations of US dispersion of the ZnO suspension.

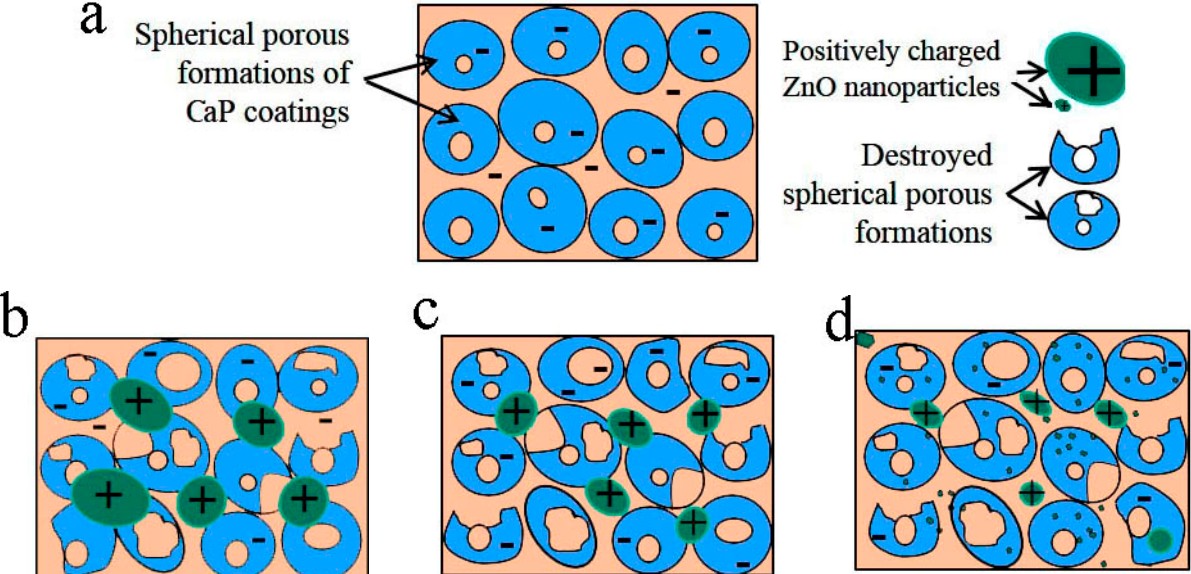

**Figure 9.** Scheme of the electrical charge distribution on the surface of MAO coatings without ZnO nanoparticles (**a**) and after different durations of US dispersion: 20 min (**b**), 40 min (**c**), 60 min (**d**).

The scheme illustrates the character of ZnO distribution on the coating surface. The C-P coating without ZnO nanoparticles had numerous spherical porous formations on its surface and a negatively charged surface.

The morphology of the initial coating comprised undestroyed spherical porous formations (Figure 9a). Destroyed spherical formations and their fragments were observed in the coatings after 20 min of US dispersion of the ZnO suspension due to ultrasonic

vibrations (Figure 9a, marked with black arrows). In addition, positively charged ZnO agglomerates were observed on the negatively charged surface of the coatings. With an increase in the US dispersion duration from 20 to 60 min, the surface relief became rougher due to destroyed formations; additionally, the ZnO agglomerates became smaller and distributed more homogeneously. This led to an increase in the roughness parameter (Ra) of the coatings from 3.4 to 5.0 μm.

### 3.4. Electrochemical Properties of Modified Coatings

An investigation of the electrochemical properties of the initial coating and the coatings modified with ZnO nanoparticles was performed. The analysis of the potentiodynamic polarization curves (Figure 10) shows that the curves are located close together, which indicates an insignificant difference in the corrosion resistance of the samples.

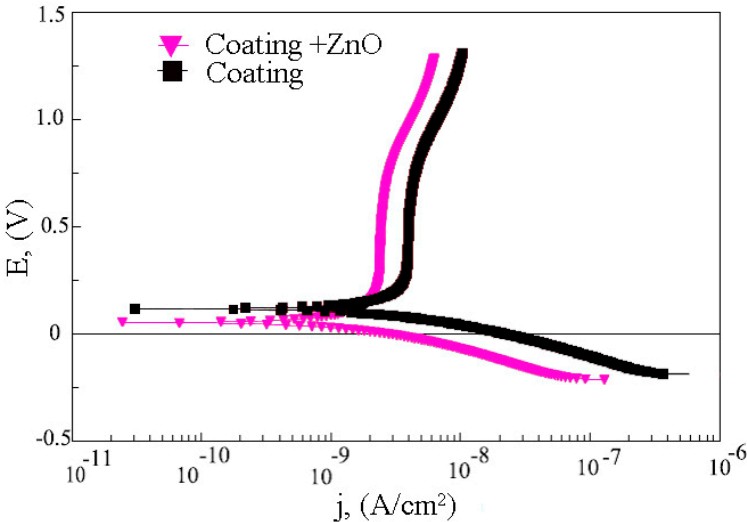

**Figure 10.** Potentiodynamic polarization curves of C-P coating before and after modification of coatings with ZnO nanoparticles after 60 min of US dispersion.

It was revealed that the corrosion resistance of the coating modified with ZnO nanoparticles was higher than that of the initial coating. The coating resistance and impedance module increased from $2.4 \cdot 10^6$ to $3.1 \cdot 10^6$ $\Omega \cdot cm^2$ and from $3.7 \cdot 10^6$ to $5.8 \cdot 10^6$, respectively (Table 2). Moreover, the addition of ZnO nanoparticles to the C-P coating led to a decrease in the corrosion current density of the coatings from $2.5 \cdot 10^{-9}$ to $1.5 \cdot 10^{-9}$ $A/cm^2$. This indicates an increase in the corrosion resistance of the coatings after modification with ZnO nanoparticles.

**Table 2.** Electrochemical parameters of the samples with C-P coatings.

| Samples | Corrosion Potential (E), V | Corrosion Current Density (I), A/cm$^2$ | Polarization Resistance (Rp), $\Omega \cdot cm^2$ | Impedance Modulus (⏐Z⏐), $\Omega \cdot cm^2$ |
|---|---|---|---|---|
| CP coatings | 0.12 | $2.5 \cdot 10^{-9}$ | $2.4 \cdot 10^6$ | $3.7 \cdot 10^6$ |
| CP coatings with ZnO | 0.05 | $1.5 \cdot 10^{-9}$ | $3.1 \cdot 10^6$ | $5.8 \cdot 10^6$ |

### 3.5. Biological Properties of Modified Coatings

The MTT study of the coating extracts on L929 cell viability showed that the fractions of surviving cells with the unmodified coating and the coating with ZnO nanoparticles were equal to 93% and 85%, respectively (Figure 11). The introduction of ZnO nanoparticles led to a slight decrease in viable cells. However, such coatings are not toxic and are classified as materials with a low degree of toxicity (ISO 10993-5: 2009). Moreover, the cytotoxicity of ZnO compounds on human cells has already been studied [36]. Cells exhibited up to 80%

cell survival after incubation with ZnO particles, which was assessed by the authors as a high level of viability. Thus, the coatings obtained in this work should not have a cytotoxic effect on cells.

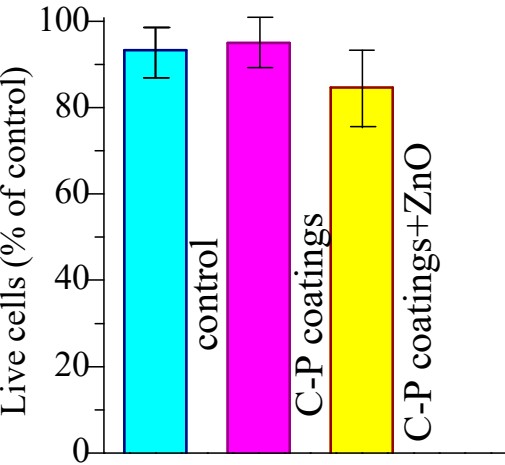

**Figure 11.** Viability indicators of L929 fibroblasts when incubated with extracts of the C-P coatings.

The antibacterial activity of samples towards Gram-positive MRSA was evaluated via the suspension approach. Figure 12 shows images of Petri dishes with *Staphylococcus aureus*, obtained after 3 and 6 h of incubation on the surface of samples covered with C-P coatings. The C-P coatings with ZnO were found to have greater antibacterial activity than the C-P coating. The number of colony-forming units (CFU) per 100 mL on unmodified C-P coatings decreased only from 8700 down to 7440 (Figure 12, Table 2). The number of CFU/100 mL on the C-P coatings modified with ZnO nanoparticles strongly decreased, as compared to that on the initial coatings, after 3 and 6 h incubation, as shown in Figures 12c,f and 13. The results showed that the C-P coating had a weak effect on MRSA, whereas there was an apparent effect of ZnO-NPs against the same strain.

Thus, zinc oxide nanoparticles, entering the coating through the open pore system, gave it an antibacterial effect. In the body, as the coating dissolves, the antibacterial agents will be released locally. ZnO nanoparticles are widely used due to their antimicrobial properties and are highly biocompatible. But their main mechanism of antibacterial action has not been fully elucidated. Electrostatic interaction, $Zn^{2+}$ ion release, and the generation of reactive oxygen species (ROS) have been described as the main pathways of the antibacterial activity of ZnO NPs [37]. Zinc oxide is an n-type semiconductor with high binding energy [38]. ZnO nanoparticles can interact with water to form ROS [39]. The ROS can destroy cell membranes, proteins, and DNA through oxidation and eventually cause bacterial inactivation. Another antibacterial mechanism occurs through the release of zinc ions ($Zn^{2+}$) that damage the cell membrane, interrupting some metabolic pathways. In addition, bacterial membrane teichoic and lipoteichoic acids act as a chelating agent on $Zn^{2+}$ ions, which are then carried by passive diffusion across membrane proteins and destroy them [40]. Thus, additional studies can relevantly contribute to the prediction of possible mechanisms of ZnO NP antibacterial action.

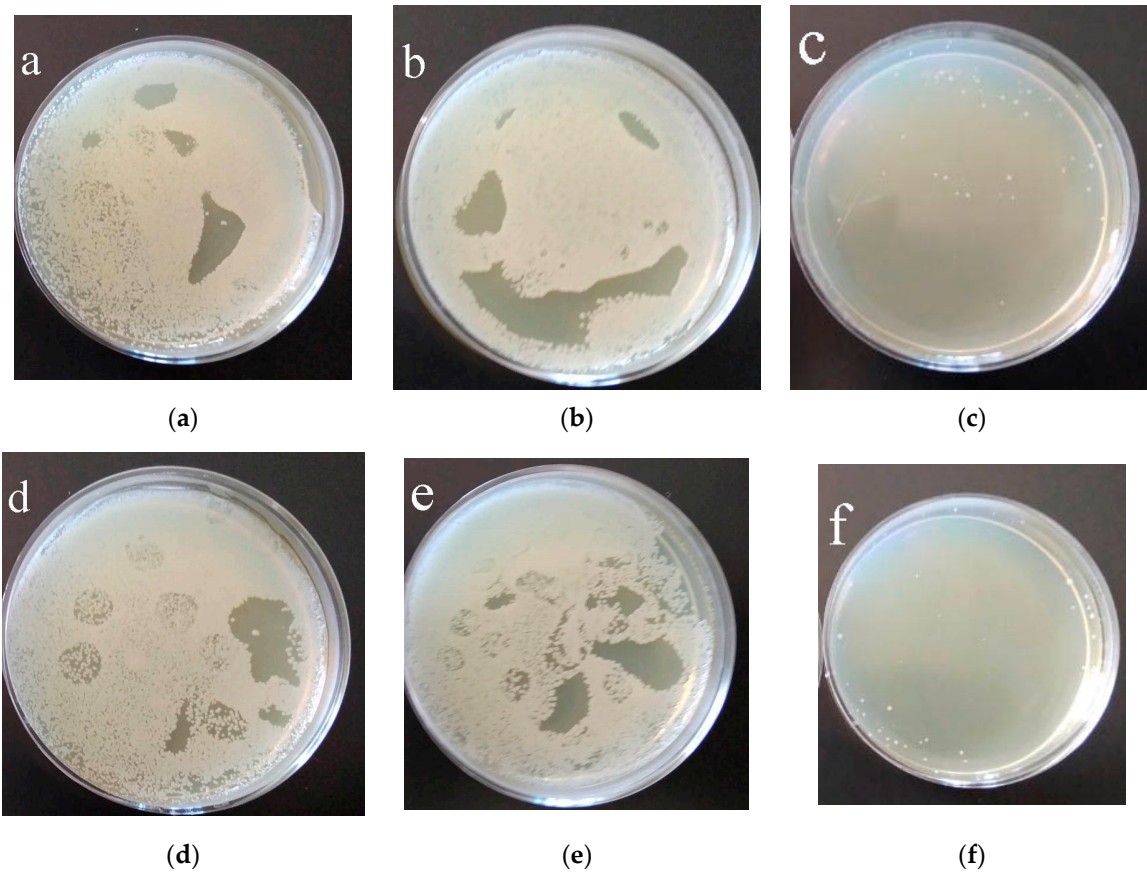

**Figure 12.** Results of 3 h (**a–c**) and 6 h (**d–f**) growth of MRSA in agar medium: (**a,d**) control 1; (**b,e**) C-P coating; (**c,f**) C-P coating modified with ZnO nanoparticles after 60 min of US dispersion of ZnO suspension.

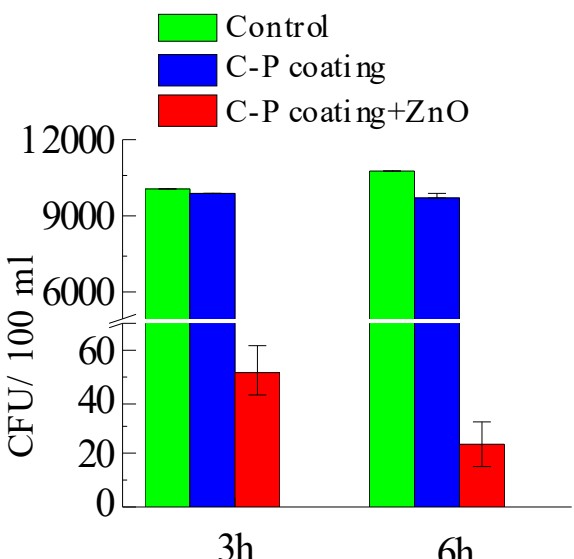

**Figure 13.** Results of 3 and 6 h growth of MRSA in agar medium.

## 4. Conclusions

Bioactive bactericidal calcium phosphate coatings with ZnO nanoparticles were obtained via a micro-arc oxidation process. The phase composition of all coatings was represented by amorphous calcium phosphates and a few peaks of crystalline ZnO and Ti

(substrate). The zeta potential of the C-P coating with ZnO nanoparticles linearly increased from −60 mV to −53 mV through partial charge compensation. The corrosion resistance of the coating modified with ZnO suspension after 60 min of US dispersion was $3.1 \cdot 10^6$ $\Omega \cdot$cm$^2$. The modification of the coatings with ZnO nanoparticles allowed us to increase the anticorrosion characteristics by about 30%. Thus, the calcium phosphate coating modified with ZnO nanoparticles after 60 min of US dispersion of the ZnO suspension had a homogeneous morphology and a high roughness, porosity, zeta potential value, and corrosion properties.

The coatings with ZnO nanoparticles showed a slight decrease in fibroblast cell line L929. Coatings with ZnO are classified as materials with a low degree of toxicity. The antibacterial activity of the coatings with the addition of ZnO nanoparticles against MRSA No. 209 was 99%, exceeding the antibacterial activity of the initial C-P coating.

The introduction of ZnO nanoparticles into a calcium phosphate coating helps to increase its antibacterial properties. An implant with such a coating will be less susceptible to infection and have a lower risk of rejection, but to recommend this coating for bone implants, further research is required.

**Author Contributions:** V.C., conceptualization, methodology, investigation, writing—original draft preparation, visualization, project administration; M.S., methodology, investigation, formal analysis; O.B. and M.K., formal analysis, methodology, investigation, data curation, visualization; A.M. and A.T., methodology, formal analysis, investigation; K.G. and A.Z. formal analysis, validation, methodology, investigation; V.E. and I.V. formal analysis, validation, methodology, investigation; Y.S., formal analysis, methodology, investigation, data curation. All authors have read and agreed to the published version of the manuscript.

**Funding:** The work was performed according to the Government research assignment for ISPMS SB RAS, project FWRW-2021-0007.

**Institutional Review Board Statement:** Not applicable.

**Informed Consent Statement:** Not applicable.

**Data Availability Statement:** Data are contained within the article.

**Acknowledgments:** This research was supported in part by Siberian State Medical University development program Priority 2030.

**Conflicts of Interest:** The authors declare that they have no conflict of interest to report regarding the present study.

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
