# Peer review of "Antibacterial Calcium Phosphate Coatings with Zinc Oxide Nanoparticles"

_coatings, doi:10.3390/coatings13111921_

Round 1
Reviewer 1 Report
Comments and Suggestions for Authors
General comment – the approach is interesting and appears to be promising. The MAO method is not one that I’ve seen used to put hydroxyapatite on actual medical implants and I think the paper would benefit from a more thorough introduction. For example, is there prior literature that should be included describing the CaP coatings produced by this method? The literature review could benefit from a focused description of the CaP deposited by MAO and less focus on other materials.
The ZnO particles are clearly deposited onto the CaP surface, but is there any indication that the particles are adhered in any way? Would they simply shake loose if the coated parts were shipped? Or is there an electrical adhesion mechanism?
Overall, I think that this is a relatively new approach to deposit CaP coatings and it would have helped if the authors could add in some comparisons with the blank metal and not just use the CaP coating as a single control surface. That would allow a more effective judgement on the suitability of this approach.
The Materials and Methods section needs a lot of work. It is far too light and provides nowhere near enough information to allow the coating to be replicated.
I have also included some specific queries below and if the authors can consider these, then I think it will be worthy of publication.
Specific comments:
Line 27. CoCr is still a significant option for orthopaedic and dental implants and is also very widely used. It’s not just Ti based.
Line 33 – 35. That’s one way but it’s not the only way.
Line 46. Add a reference to verify this.
Lines 63 – 67. The authors state that the coatings were made from stoichiometric hydroxyapatite and also from powders of CaCO3 and H3PO4. Which one is true?
Line 72-74. What was the concentration of ZnO in the 25 ml? What power level and frequency was used for the ultrasonics? Was the ZnO dispersion placed directly into the bath or into a container that was then immersed in the bath? If so, what type of container and did this impact the ulstrasonic energy transferred into the ZnO dispersion.
Line 101 – Samples were immersed with the cells, but how were they evaluated afterwards? Were the cells in solution counted or the cells on the surface? There’s no where near enough information to allow for replication of the experiment.
Line 106 – 105 cfu/ml is a very dilute amount of bacteria and it’s a very specific number. Should that be 105 instead?
Lines 102 – 109. How long was the bacteria exposed to the surface for? At what temperature? How were bacteria recovered from the surface? How were the cells counted – was it in solution or on the surface?
Figure 2 – are there error bars for this because it looks as if the particle size may not have changed all that much and I’m not sure the change is statistically meaningful.
Lines 124 – 129. I don’t understand what the authors are trying to communicate here.
Figure 3 caption – (b) – (h). Are these times? Or energy levels? Or something else?
Figure 3 – why not include an image of the uncoated metal for comparison.
Table 1 – why was the Ca/P ratio so far off the idealised ratio of hydroxyapatite. Shouldn’t there by more Ca than P? Why does the Zn concentration decrease with treatment time?
Line 192 – 193 – has the material been removed from the coating?
Line 244 – there is no mention of an MTT assay method in the experimental section. Why were fibroblasts chosen for this study? Surely osteoblasts would be the relevant choice for a bone fixation study.
Lines 244 – 248 – There may be a mild cytotoxic effect evident there. Do CaP coatings normally increase cell attachment and proliferation? The surface was roughened by the coating and this should have promoted cell adhesion and proliferation. But that didn’t happen. Instead, the cell numbers decreased.
Line 260 – what is a KOE and why do the authors not use the standard cfu metric?
Figure 13 – what is the EIE/100 ml measurement? I don’t know what that represents. Is that a measure of cfu? I think these might be good results, but I’m sure.
Line 266 – 267 – The claim to give gradual release lacks any supporting data.
Conclusions – Lines 283 – 284. The cytotoxicity claim is dubious, as outlined above.
Comments on the Quality of English Language
Some of the English can difficult to follow and the manuscript would benefit from some minor updates.
Author Response
General comment – the approach is interesting and appears to be promising. The MAO method is not one that I’ve seen used to put hydroxyapatite on actual medical implants and I think the paper would benefit from a more thorough introduction. For example, is there prior literature that should be included describing the CaP coatings produced by this method? The literature review could benefit from a focused description of the CaP deposited by MAO and less focus on other materials.
The ZnO particles are clearly deposited onto the CaP surface, but is there any indication that the particles are adhered in any way? Would they simply shake loose if the coated parts were shipped? Or is there an electrical adhesion mechanism?
Response: During our studies of the coatings’ properties we implicitly confirmed the adhesion of nanoparticles to the coating. After 1 day keeping the coatings in solution with fibroblasts cells monolayer there still were assemblies of ZnO nanoparticles as well as elemental analysis confirmed those presence in the coating (fig. below). Moreover, in the manuscript we confirmed the fact that nanoparticles penetrated into coatings through the pores. The mechanism of nanoparticles adhesion to the coating is understudied and is still imperfectly clear but it is most likely that you are right and there is an electrical adhesion mechanism.
Overall, I think that this is a relatively new approach to deposit CaP coatings and it would have helped if the authors could add in some comparisons with the blank metal and not just use the CaP coating as a single control surface. That would allow a more effective judgement on the suitability of this approach.
The Materials and Methods section needs a lot of work. It is far too light and provides nowhere near enough information to allow the coating to be replicated.
Response: Thank you for your comment. The authors added information to Materials and Methods section.
I have also included some specific queries below and if the authors can consider these, then I think it will be worthy of publication.
Specific comments:
- Line 27. CoCr is still a significant option for orthopaedic and dental implants and is also very widely used. It’s not just Ti based.
Response: The authors have added the appropriate information to the introduction.
- Line 33 – 35. That’s one way but it’s not the only way.
Response: The authors have added the information to the introduction.
- Line 46. Add a reference to verify this.
Response: Reference was added
- Lines 63 – 67. The authors state that the coatings were made from stoichiometric hydroxyapatite and also from powders of CaCO3 and H3PO4. Which one is true?
Response: To obtain calcium-phosphate coating it is necessary to use all the three compounds. Hydroxyapatite is added into the water solution of ortho-phosphoric acid, which is required for the formation of oxide layer during the deposition of coating on Ti surface. Calcium carbonate (СаСО3) is added into the electrolyte for the increasing calcium concentration as a result of its reaction with H3PO4 acid.
- Line 72-74. What was the concentration of ZnO in the 25 ml? What power level and frequency was used for the ultrasonics? Was the ZnO dispersion placed directly into the bath or into a container that was then immersed in the bath? If so, what type of container and did this impact the ulstrasonic energy transferred into the ZnO dispersion.
Response: In order to make a suspension we used 30 mg per 25 ml of distilled water. Such ratio powder to fluid was empirically chosen. Ultrasonic bath’s power was 100 W and operating frequency was 35 kHz (added into the text). ZnO suspensions were placed into the bath inside the glass tumbler as it is shown in fig. 1. Using of glass tumblers dos not resist the ultrasonic treatment of ZnO powder suspension.
- Line 101 – Samples were immersed with the cells, but how were they evaluated afterwards? Were the cells in solution counted or the cells on the surface? There’s no where near enough information to allow for replication of the experiment.
Response: We added in Materials and Methods part (Page 3, line 124, marked yellow):
«The cytotoxicity of the C-P coatings was studied by MTT test using the mouse fibroblast L929 cell line (Russian Research Center of Virology and Biotechnology VECTOR, Russia). This cell line is recommended by ISO 10993-5 due to its high sensitivity to toxic effects. The cells were cultured within 24 h in Dulbecco's Modified Eagle Medium (HiClone, USA) with the addition of 10% fetal bovine serum (HiClone, USA) and 1% penicillin/streptomycin (Biolot, Russia) at 37 °C and 5% CO2 (Sanyo, Japan). For cytotoxicity investigation by extract test, test samples with 2 ml DMEM media were placed in the wells of a 24-well plate and incubation for 24 hours at 37 °C. The samples were taken out and a cell suspension was added in the amount of 50 000 cells per sample. The cells were incubated for 24 hours at 37 °C in an atmosphere of 5% CO2. After incubation, cells were detached from the surface of the composite samples and transferred 100 μL cell suspensions each into 96-well plates. Then 10 µL of MTT solution (3-(4.5-dimethyl thiazol-2-yl)-2.5-diphenyl-tetrazolium bromide) was added to each well with cells. Incubation with MTT solution was at 37±1 °C for 2 hours in an atmosphere of 5% CO2. At the end of incubation, the nutrient medium was carefully removed and 100 μL of dimethyl sulfoxide (Biolot, Russia) was added to each well to dissolve formazan crystals. After 15 minutes, the optical density was determined on a Multiscan FC microplate spectrophotometer (Termo Scientific, Germany) at a wavelength of 570 nm. Then the percentage of living cells (viability, %) was calculated».
- Line 106 – 105 cfu/ml is a very dilute amount of bacteria and it’s a very specific number. Should that be 105instead?
Response: Thank you for pointing these out. This mistake was corrected.
- Lines 102 – 109. How long was the bacteria exposed to the surface for? At what temperature? How were bacteria recovered from the surface? How were the cells counted – was it in solution or on the surface?
Response: We added in Materials and Methods part (Page 4, line 143, marked yellow):
«The antibacterial activity of the C-P coating samples was determined by counting the residual bacteria using the CFUs counting method against methicillin resistance Staphylococcus aureus (MRSA) hospital strain No. 209. MRSA was cultured at 37 °C in Mueller–Hinton agar (Merck, USA) for 24 hours. The test samples were placed in Petri dishes with a diameter of 60 mm. A volume of 1 µL per 1 mm2 sample area of the bacterial suspension in Mueller-Hinton broth (HiMedia Laboratories, India) with a concentration 105 СFU/mL was placed to the samples surface and spread evenly on the surface using a sterile swab. The exposure time was 3 and 6 hours at 37 °C and a humidity of 90%. The residual number of viable bacteria (CFU/ml) was counted after 3 and 6 hours of incubation at 37 °C. After time of exposure, samples were placed in sterile containers containing 10 mL of neutralizing agent (PBS, HiMedia Laboratories, India) and shaken for 10 min using an orbital shaker PSU-20i (SIA Biosan, Latvia) to remove adhered bacteria. From each supernatant, 100 µL were placed onto Mueller-Hinton agar plates. The prepared plates were subsequently incubated at 37±1 °C for 24 h. The colony plate count method was used for the enumeration of CFUs after incubation. The antibacterial rate (AR) of each sample was evaluated by calculating the percent (%) reduction in microbial contamination of test samples compared to the control samples. The bacteria in PBS were used only as control samples.»
- Figure 2 – are there error bars for this because it looks as if the particle size may not have changed all that much and I’m not sure the change is statistically meaningful.
Response: Error bars were added in fig. 2.
- Lines 124 – 129. I don’t understand what the authors are trying to communicate here.
Response: Provided refer contains images of AlOOH nanoparticles’ assembly after hydrolysis (Fig. 3e). The text compares two kinds of nanoparticles as well as their shape. More complicated shape of AlOOH nanoparticles leads to the strong agglomeration of nanoparticles as compared to ZnO nanoparticles. Due to this fraction of ZnO particles with minimal size of 20-50 nm is significantly more than fraction of particles of larger size. This one can explain by the shape of ZnO nanoparticles.
Figure 3 e
For better illustration we added TEM-images of AlOOH nanoparticles after hydrolysis and ZnO. Also, in text we added the reference to the published article containing those images.
Figure 2. TEM of aluminum oxide particles obtained by oxidation in water [Antitumor activity of low-dimensional alumina structures M. S. Korovin and A. N. Fomenko]
Figure 2. TEM image of ZnO nanopowder
- Figure 3 caption – (b) – (h). Are these times? Or energy levels? Or something else?
Response: There are US dispersion of ZnO suspension of different duration. The appropriate information was added to the figure caption.
- Figure 3 – why not include an image of the uncoated metal for comparison.
Response: The manuscript was aimed to estimate the nanoparticles’ influence on the properties of coatings. Comparison of uncoated and coated metal’s properties was not carried out. But to be clearer we added the image of uncoated metal’s surface.
- Table 1 – why was the Ca/P ratio so far off the idealised ratio of hydroxyapatite. Shouldn’t there by more Ca than P? Why does the Zn concentration decrease with treatment time?
Response: Anodizing in an acidic environment leads to a partial penetration of titanium into the coating, which slightly reduces the ratio of elements. Moreover, a positive potential was applied to the substrate, which led to the deposition of negatively charged electrolyte ions (figure). After a coating with a predominant negative charge was formed on titanium, positively charged electrolyte ions also begin to be attracted and deposited on the coating. But their number is less than negative ions.
Schematic illustration of possible compounds in the electrolyte based on hydroxyapatite
Also, it should be noted that the lower the Ca/P ratio, the faster the coatings dissolve and are replaced by bone tissue, the cells of which are attached to the rough porous surface.
The amount of zinc decreases with increasing dispersion duration as the size of agglomerates with ZnO nanoparticles attached to the coating decreases. Hence, less zinc is deposited on the surface, and smaller particles (the number of which increases with increasing dispersion duration) penetrate deeper into the pores of the coating. And elemental analysis was carried out only on the surface of the coating.
- Line 192 – 193 – has the material been removed from the coating?
Response: As can be seen in Figures 3 and 5, there is partial destruction of the spherical structures of the coating, but the main coating material remains on the substrate. The thinnest walls of the coating spheres on the surface are destroyed and the coating material is partially lost. But judging by SEM images of the coating surface, the damage is minimal.
- Line 244 – there is no mention of an MTT assay method in the experimental section. Why were fibroblasts chosen for this study? Surely osteoblasts would be the relevant choice for a bone fixation study.
Response: We added in Materials and Methods part (Page 3, line 124, marked yellow):
«The cytotoxicity of the C-P coatings was studied by MTT test using the mouse fibroblast L929 cell line (Russian Research Center of Virology and Biotechnology VECTOR, Russia). This cell line is recommended by ISO 10993-5 due to its high sensitivity to toxic effects.»
- Lines 244 – 248 – There may be a mild cytotoxic effect evident there. Do CaP coatings normally increase cell attachment and proliferation? The surface was roughened by the coating and this should have promoted cell adhesion and proliferation. But that didn’t happen. Instead, the cell numbers decreased.
Response: The amount of living cells decreased slightly and is within the error limits. Which indicates only the absence of a toxic effect of calcium phosphate coating. To assess the proliferation of cells and their growth on the surface of the coating, an additional study needs to be carried out, which is planned in the future.
- Line 260 – what is a KOE and why do the authors not use the standard cfu metric?
Response: KOE is the Russian meaning CFU. The term KOE was corrected and replaced with CFU.
- Figure 13 – what is the EIE/100 ml measurement? I don’t know what that represents. Is that a measure of cfu? I think these might be good results, but I’m sure.
Response: It is CFU. KOE is the Russian meaning CFU. The term KOE was corrected and replaced with CFU.
- Line 266 – 267 – The claim to give gradual release lacks any supporting data.
Response: We agree with the comment. We have slightly adjusted the sentence in the text.
- Conclusions – Lines 283 – 284. The cytotoxicity claim is dubious, as outlined above.
Response: We added in Conclusion part (Page 14, line 366, marked yellow):
«The coatings showed weak toxic effect against sensitive fibroblast cell line L929. The more than 80% cell viability was observed after 24 hours incubated with sample».

Reviewer 2 Report
Comments and Suggestions for Authors
This manuscript presents a detailed study of bactericidal calcium phosphate coatings with the incorporation of zinc oxide (ZnO) nanoparticles. These coatings were fabricated through the micro arc oxidation method on a titanium substrate, aiming to enhance their antibacterial properties. The ZnO nanoparticles were integrated into the calcium phosphate (C-P) coatings using impregnation and ultrasonic (US) dispersion for various durations ranging from 10 to 60 minutes. The physical, electrochemical, and electret characteristics of the modified coatings were assessed and compared to the unmodified ones. Additionally, biological evaluations were conducted to determine cytotoxicity on fibroblast cells and the antibacterial effectiveness against Staphylococcus aureus 209P. However, it has some correction that need attention:
1. While the manuscript uses units consistently, it would be beneficial to use International System (SI) units throughout the text for uniformity and broader accessibility.
2. The manuscript could benefit from additional details in some sections to provide a more comprehensive understanding of the research. For example, a more detailed description of the micro arc oxidation process and the specific preparation of ZnO nanoparticles could enhance clarity.
3. Proper citation and referencing are crucial in scientific manuscripts. Ensure that all sources are appropriately cited and listed in the references section. Additionally, the manuscript should adhere to a specific citation style (e.g., APA, IEEE) for consistency.
4. The data presentation, particularly in the conclusion section, could be improved. Some of the numerical values and measurements could be presented in a more structured and organized manner.
5. While the manuscript mentions the use of scanning electron microscopy and X-ray diffraction, actual figures, diagrams, or images representing the experimental results are not included in the text. The inclusion of visual elements could significantly enhance the manuscript's clarity and impact.
6. The manuscript could further emphasize the broader significance of the research findings and its potential impact in the field of bone implants and related applications.
Comments on the Quality of English LanguageThe manuscript contains several grammatical errors and awkward sentence structures that can make it challenging to read and understand. For instance, in the sentence, "This is due to destruction of spherical formations of the coating and more homogeneous ZnO nanoparticles distribution," the phrasing could be clearer.
Author Response
This manuscript presents a detailed study of bactericidal calcium phosphate coatings with the incorporation of zinc oxide (ZnO) nanoparticles. These coatings were fabricated through the micro arc oxidation method on a titanium substrate, aiming to enhance their antibacterial properties. The ZnO nanoparticles were integrated into the calcium phosphate (C-P) coatings using impregnation and ultrasonic (US) dispersion for various durations ranging from 10 to 60 minutes. The physical, electrochemical, and electret characteristics of the modified coatings were assessed and compared to the unmodified ones. Additionally, biological evaluations were conducted to determine cytotoxicity on fibroblast cells and the antibacterial effectiveness against Staphylococcus aureus 209P. However, it has some correction that need attention:
- While the manuscript uses units consistently, it would be beneficial to use International System (SI) units throughout the text for uniformity and broader accessibility.
Response: The authors have considered the comment
- The manuscript could benefit from additional details in some sections to provide a more comprehensive understanding of the research. For example, a more detailed description of the micro arc oxidation process and the specific preparation of ZnO nanoparticles could enhance clarity.
Response: Thank you for this valuable comment. The authors added information.
- Proper citation and referencing are crucial in scientific manuscripts. Ensure that all sources are appropriately cited and listed in the references section. Additionally, the manuscript should adhere to a specific citation style (e.g., APA, IEEE) for consistency.
Response: Thank you for the comment, but references were designed in accordance with the requirements of the journal.
- The data presentation, particularly in the conclusion section, could be improved. Some of the numerical values and measurements could be presented in a more structured and organized manner.
Response: The authors have corrected conclusion.
- While the manuscript mentions the use of scanning electron microscopy and X-ray diffraction, actual figures, diagrams, or images representing the experimental results are not included in the text. The inclusion of visual elements could significantly enhance the manuscript's clarity and impact.
Response: The results of the studies of the above methods are presented in Figures 2, 3, 5, 6, 7.
- The manuscript could further emphasize the broader significance of the research findings and its potential impact in the field of bone implants and related applications.
Response: Thank you for pointing these out. The authors have added information.
Thanks for the comments/suggestions on the submission. We have made the corresponding modifications to the previous manuscript. A revised manuscript with the changes has been uploaded to the paper submission system. Here below are our point-to-point responses to the comments received.

Reviewer 3 Report
Comments and Suggestions for Authors
Chebodaeva et al. fabricated porous calcium phosphate coatings with ZnO nanoparticle through micro arc oxidation method at the titanium substrate. This Manuscript is written well however many major/minor issues need to solve. My comment are as follows:
1. In abstract need to highlight the problems of the study and insert some important numerical results.
2. What is the impregnation and assisted ultrasonic (US) dispersion method novelty of this study?
3. Line 17, electret properties? check it?
4. Bacteria name should be italics. What is the size of nanoparticles?
5. Line 26. 10 ref for supporting one sentence its so strange please delete old references just keep max. 5 ref. Same also in line 29and 35. Check this strange issues throughout the manuscript.
6. Line 57-59, need to rewrite what exactly authors presented need to mention more details.
7. In introduction authors need to present some related to this study and highlight it.
8. Novelty can be rewritten better way.
9. ZnO nanoparticles is commercial if so need to mention details in the materials section. make a separate section of materials and include all materials.
10. Figure 1 resolution is low and simple need to present better way.
11. Section 2.2 title need to change to Characterizations.
12. Line 79 two common bracket.
13. Section 2.2 sample analysis parameters of each instrument need to mention.
14. Line 117, Authors written particle size 50±20 nm, the SD seems almost 50 % of mean values which is not acceptable.
15. Line 125, it should be Al(OH)2.
16. (hexagon, circle, square, rectangle), line 128, what about your particle shape and why authors need to discuss critically.
17. Figure 2 b is it Gaussian plot, if yes its is wrong.
18. Subfigure level such as a, b,…should be inside the figure , a top corner. Please arrange it.
19. Line 136, many ref just keep two.
20. Line 140, The roughness (Ra) of such coating were 3.4 μm (Fig. 4a). how authors determine need to mention in the manuscript.
21. Fig. 3 sub figures arrange in a good way. At present it is scatter.
22. Fig 4c porosity how authors calculated no information.
23. Fig 7 need to draw scientifically y axis is missing. How about only ZnO NP XRD pattern.
24. Line 206 to 215 may paragraphs make it arrangement in a right way. One two sentences should be one paragraph. Merge it accordingly.
25. Figure 9 is interesting make it color and attractive for readers.
26. Line 252, make italics of Staphylococcus aureus words. Check throughout the manuscript.
27. What is IC 50 value of antibacterial test.
28. Antibacterial test mechanism is missing.
29. Fig. 13, CFU value?
30. Antibacterial test results are poorly written no information.
31. Conclusion have many paragraphs and too much text. Make it short preciously with one paragraph.
Comments on the Quality of English LanguageMinor editing of English language required
Author Response
Chebodaeva et al. fabricated porous calcium phosphate coatings with ZnO nanoparticle through micro arc oxidation method at the titanium substrate. This Manuscript is written well however many major/minor issues need to solve. My comment are as follows:
- In abstract need to highlight the problems of the study and insert some important numerical results.
Response: We have added information to the abstract. An explanation of the research problems is given and expanded in the introduction.
- What is the impregnation and assisted ultrasonic (US) dispersion method novelty of this study?
Response: The novelty lies in the post-modification of the formed coatings without significant changes in the elemental and phase composition of the coating. Moreover, the developed surface relief of the coating and porosity were also preserved after such modification with nanoparticles. At the same time, it was possible to change such important characteristics as the zeta potential of the coating and its antibacterial properties.
- Line 17, electret properties? check it?
Response: Thank you for your comment. The authors have corrected term «electret».
- Bacteria name should be italics. What is the size of nanoparticles?
Response: The words styles were corrected. The average ZnO nanoparticle size was 50±2 nm.
- Line 26. 10 ref for supporting one sentence its so strange please delete old references just keep
max. 5 ref. Same also in line 29and 35. Check this strange issues throughout the manuscript.
Response: Thank you for pointing these out. These links have been checked and redone.
- Line 57-59, need to rewrite what exactly authors presented need to mention more details.
Response: The authors rewrote this sentence
- In introduction authors need to present some related to this study and highlight it.
Response: Thanks for the recommendation. The authors have added information to the introduction section.
- Novelty can be rewritten better way.
Response: The authors added information about the novelty of the study.
- ZnO nanoparticles is commercial if so need to mention details in the materials section. make a separate section of materials and include all materials.
Response: ZnO nanoparticles were obtained by co-authors at the Institute of Strength Physics and Materials Science of the Siberian Branch of the Russian Academy of Sciences using the method of electric explosion of wires. Information added to section 2.1.
- Figure 1 resolution is low and simple need to present better way.
Response: Figure 1 was changed.
- Section 2.2 title need to change to Characterizations.
Response: Section 2.2 title was renamed.
- Line 79 two common bracket.
Response: Recommendation is taken into account.
- Section 2.2 sample analysis parameters of each instrument need to mention.
Response: Sample analysis parameters were added
- Line 117, Authors written particle size 50±20 nm, the SD seems almost 50 % of mean values which is not acceptable.
Response: Thank you for pointing these out. It should be 50±2 nm. The nanoparticles average size was corrected and replaced in the text.
- Line 125, it should be Al(OH)2.
Response: AlO(OH) is an aluminum oxyhydroxide compound, known as boehmite. And Al(OH)2 - aluminum hydroxide
- (hexagon, circle, square, rectangle), line 128, what about your particle shape and why authors need to discuss critically.
Response: Figure 2 shows TEM images of our ZnO nanoparticles and shows different shapes of nanoparticles (hexagon, circle, square, rectangle). This was provided in order to highlight the influence of the shape of the nanoparticles used on the size of agglomerates formed in suspensions.
For better illustration, we added a TEM image of AlO(OH) nanoparticles after hydrolysis and ZnO, and in the text we will add a reference to the article with these images.
Figure 2. TEM of aluminum oxide particles obtained by oxidation in water [Antitumor activity of low-dimensional alumina structures M. S. Korovin and A. N. Fomenko]
Figure 2. TEM image of ZnO nanopowder
- Figure 2 b is it Gaussian plot, if yes its is wrong.
Response: No, this is not a Gaussian plot. These lines are drawn to demonstrate a tendency towards an increase in the proportion of small ZnO particles.
- Subfigure level such as a, b,…should be inside the figure , a top corner. Please arrange it.
Response: Authors corrected the subfigure level.
- Line 136, many ref just keep two.
Response: The authors reduced the number of references.
- Line 140, The roughness (Ra) of such coating were 3.4 μm (Fig. 4a). how authors determine need to mention in the manuscript.
Response: Coatings’ roughness is an important surface property affecting the interaction with cells. Provide the roughness of the primary coating without nanoparticles was necessary to assess the effect of further modification on the roughness of the coatings.
- 3 sub figures arrange in a good way. At present it is scatter.
Response: Figure 3 shows typical images for each duration of ultrasonic dispersion of ZnO suspensions. For each sample at least 10 images were obtained to confirm the pattern of decreasing sizes of ZnO agglomerates and increasing the proportion of smaller ZnO agglomerates.
- Fig 4c porosity how authors calculated no information.
Response: Thank you for your comment. The authors have added information to Materials and Methods section.
- Fig 7 need to draw scientifically y axis is missing. How about only ZnO NP XRD pattern.
Response: Thank you for comment. The authors corrected XRD pattern. For the purpose of this article, it is important to compare coatings with ZnO nanoparticles at the required duration of dispersion with unmodified coatings.
- Line 206 to 215 may paragraphs make it arrangement in a right way. One two sentences should be one paragraph. Merge it accordingly.
Response: Thank you for pointing these out. It was corrected.
- Figure 9 is interesting make it color and attractive for readers.
Response: Thank you for valuable comment. The Figure 9 was colored in.
- Line 252, make italics of Staphylococcus aureus words. Check throughout the manuscript.
Response: Thank you for pointing these out. The words styles were corrected.
- What is IC 50 value of antibacterial test.
Response: In the antibacterial test, the IC 50 was not determined because we used only one ZnO concentration. The antibacterial rate of sample was evaluated by calculating the percent (%) reduction in microbial contamination of test samples compared to the control samples.
- Antibacterial test mechanism is missing.
Response: We added in Results and Discussion part (Page 12, line XXX, marked green):
«ZnO NPs are widely used due to their antimicrobial properties and highly biocompatible. But their main mechanism of antibacterial action has not been fully elucidated. Electrostatic interaction, Zn2+ ion release and the generation of reactive oxygen species (ROS) were described as main pathways of the antibacterial activity of ZnO NPs [Mendes, C. R., Dilarri, G., Forsan, C. F., Sapata, V. D. M. R., Lopes, P. R. M., de Moraes, P. B., ... & Bidoia, E. D. (2022). Antibacterial action and target mechanisms of zinc oxide nanoparticles against bacterial pathogens. Scientific Reports, 12(1), 2658.]. The zinc oxide is n-type semiconductor with high binding energy [Janotti, A., & Van de Walle, C. G. (2009). Fundamentals of zinc oxide as a semiconductor. Reports on progress in physics, 72(12), 126501.]. ZnO NPs can interact with water to the formation of ROS [Alaya, L., Saeedi, A. M., Alsaigh, A. A., Almalki, M. H., Alonizan, N. H., & Hjiri, M. (2023). ZnO: V nanoparticles with enhanced antimicrobial activities. Journal of Composites Science, 7(5), 190.]. The ROS can cause destroy cell membranes, proteins, and DNA through oxidation; and eventually cause bacterial inactivation. Another antibacterial mechanism occurs through the zinc ions release (Zn2+) that damage the cell membrane, interrupt some metabolic pathways. In addition bacterial membrane teichoic and lipoteichoic acids act as a chelating agent on Zn2+ ions, which are then carried by passive diffusion across membrane proteins and destroy [Lallo da Silva, B., Abuçafy, M. P., Berbel Manaia, E., Oshiro Junior, J. A., Chiari-Andréo, B. G., Pietro, R. C. R., & Chiavacci, L. A. (2019). Relationship between structure and antimicrobial activity of zinc oxide nanoparticles: An overview. International journal of nanomedicine, 9395-9410.]. Thus, additional studies can relevantly contribute to the prediction of possible mechanisms of ZnO NP antibacterial action.»
- 13, CFU value?
Response: It is CFU. The term KOE was corrected and replaced with CFU.
- Antibacterial test results are poorly written no information.
Response: We changed antibacterial activity description in Results and Discussion part (marked red)
- Conclusion have many paragraphs and too much text. Make it short preciously with one paragraph.
Response: We have shortened the text in the conclusion

Round 2
Reviewer 1 Report
Comments and Suggestions for Authors
The authors have made significant improvements to the paper and it is now much easier to understand how the experiments were carried out. The improvement in the Methods section is particularly notable.
My only residual concern is that the Conclusions do not fully agree with the statements in the body of the manuscript with regards to biocompatibility.
Section 3.5, Line 300 - 302 states that the coating was non-toxic, even though a 15% reduction in cell number was detected. The Conclusions states that there was a "weak toxic effect" (line 362). I would prefer if Section 3.5 was amended to align with the conclusions.
I would have liked to see a comment in the discussion as to why the C-P coating did not improve cell adhesion and proliferation.
Comments on the Quality of English LanguageThe standard of English is decent, but can be improved slightly. For example, line 39 - 41 states that "The effective way to create the required chemical composition and roughness of the implant surface is the formation of bioactive calcium phosphate (C-P) coating on the metal substrate" That's not the only way to achieve this goal. So the sentence should read "An effective way to .....", which is less definitive but more accurate. The manuscript has a number of issues like this.
Author Response
The authors, taking into account the reviewer's comment and previous
studies of these coatings, decided to repeat the cytotoxicity test on
several samples with a C-P coating and updated the data in the article.
However, fluctuations in values within the error compared to the control
are acceptable. Thank you for your attention to the research results.
The text of the manuscript has also been revised.
Reviewer 3 Report
Comments and Suggestions for Authors
Authors improved the manuscript.
Comments on the Quality of English LanguageMinor English need to be checked.
Author Response
The text of the manuscript has been revised.